# Development and Validation of a Prognostic Model for Esophageal Adenocarcinoma Based on Necroptosis-Related Genes

**DOI:** 10.3390/genes13122243

**Published:** 2022-11-29

**Authors:** Suhong Zhang, Shuang Liu, Zheng Lin, Juwei Zhang, Zhifeng Lin, Haiyin Fang, Zhijian Hu

**Affiliations:** 1Department of Epidemiology and Health Statistics, Fujian Provincial Key Laboratory of Environment Factors and Cancer, School of Public Health, Fujian Medical University, Fuzhou 350108, China; 2Key Laboratory of Ministry of Education for Gastrointestinal Cancer, Fujian Medical University, Fuzhou 350108, China

**Keywords:** necroptosis-related genes, esophageal adenocarcinoma, TCGA, prognostic model, bioinformatics analysis

## Abstract

Necroptosis is a newly developed cell death pathway that differs from necrosis and apoptosis; however, the potential mechanism of necroptosis-related genes in EAC and whether they are associated with the prognosis of EAC patients remain unclear. We obtained 159 NRGs from the Kyoto Encyclopedia of Genes and Genomes (KEGG) and performed differential expression analysis of the NRGs in 9 normal samples and 78 EAC tumor samples derived from The Cancer Genome Atlas (TCGA). Finally, we screened 38 differentially expressed NRGs (DE-NRGs). The results of the GO and KEGG analyses indicated that the DE-NRGs were mainly enriched in the functions and pathways associated with necroptosis. Protein interaction network (PPI) analysis revealed that TNF, CASP1, and IL-1B were the core genes of the network. A risk score model based on four DE-NRGs was constructed by Least Absolute Shrinkage and Selection Operator (LASSO) regression, and the results showed that the higher the risk score, the worse the survival. The model achieved more efficient diagnosis compared with the clinicopathological variables, with an area under the receiver operating characteristic (ROC) curve of 0.885. The prognostic value of this model was further validated using Gene Expression Omnibus (GEO) datasets. Gene set enrichment analyses (GSEA) demonstrated that several metabolism-related pathways were activated in the high-risk population. Single-sample GSEA (ssGSEA) provided further confirmation that this prognostic model was remarkably associated with the immune status of EAC patients. Finally, the nomogram map exhibited a certain prognostic prediction efficiency, with a C-index of 0.792 and good consistency. Thus, the prognostic model based on four NRGs could better predict the prognosis of EAC and help to elucidate the mechanism of necroptosis-related genes in EAC, which can provide guidance for the target prediction and clinical treatment of EAC patients.

## 1. Introduction

Esophageal cancer is a deadly malignant tumor. According to the Global Cancer Statistics 2020 [1], the incidence of esophageal cancer and its mortality rate are ranked seventh and sixth among all tumors, respectively. The incidence of EAC is increasing rapidly in many high-income countries, and this trend is expected to continue, with the incidence of EAC surpassing that of ESCC [2]. EAC patients have worse prognosis than most other types of cancer, although their prognosis has improved slightly with radiation and chemotherapy, drug therapy, and surgery. In the Western population, only 20% of patients survive for five years [3]. Given the critical role of molecularly targeted therapies in the treatment of malignancies, EAC patients need to identify effective and accurate therapeutic targets.

Apoptosis is a prevalent form of cell death in living organisms. With the in-depth study of cell death mechanisms, increasingly new cell death modalities are being identified and reported. Necroptosis is a pattern of cell death mediated by serine/threonine protein kinase 1/3 (RIPK1/RIPK3) and characterized by the activation of mixed-spectrum kinase structural domain-like proteins (MLKL/pMLKL) by phosphorylation signaling pathways [4]. This model has the morphological characteristics of necrosis-like cell death, which is characterized by lysosomal membrane degradation, cytoplasmic vacuolization, plasma membrane disassembly, and, eventually, cell explosion-like rupture [5]. Necroptosis plays a dual role in tumor progression [6]. On one hand, necroptosis inhibits tumor cell invasion, proliferation, and migration. Necroptosis factors play a regulatory role in antigen-induced T cell proliferation by eliminating excess T cells [7]. In addition to interacting directly with immune cells, by releasing DAMPs into the tumor tissue microenvironment, necroptosis factors can also initiate adaptive immune responses [8]. Yatim et al. [9] demonstrated that NF-κB activation and RIPK1 expression are critical for initiating the adaptive immunity of CD8+ T cells during programmed cell death. Feng et al. [10] showed that the overexpression of RIP3 significantly inhibited the progression of colorectal cancer. These studies suggest that necroptosis may play an antitumor role in cancer.

Necroptosis factors, on the other hand, may enhance the risk of tumor progression, possibly due to the inflammatory response triggered by necroptosis. Studies have shown that necroptosis factors can provide an inflammatory microenvironment that promotes tumor progression or an increase in reactive oxygen species, which, in turn, accelerates the malignant transformation of tumors and ultimately promotes cancer progression [11,12,13]. The mechanism by which necroptosis promotes pancreatic cancer progression may be related to the promotion of macrophage-induced adaptive immunosuppression by the CXCL1 and Mincle signaling pathways [14]. The expression of RIP3 and MLKL is upregulated in pancreatic cancer tissues, and further findings indicate that necroptosis factors can promote pancreatic cancer cell migration and invasion through the CXCL5–CXCR2 axis [15]. It can be concluded that necroptosis regulates tumorigenesis in both directions; its role in tumor development cannot be ignored, and its complex biological functions deserve further exploration. However, no systematic study has been conducted to investigate the relationship between necroptosis and EAC prognosis.

Therefore, this study was conducted to reveal the potential biological functions of differentially expressed necroptosis-related genes in EAC by exploring their expression profiles and related functional pathways. We also constructed a risk score model based on LASSO regression, and then constructed a prognostic nomogram. These results can be used to define new biomarkers with potential clinical and therapeutic relevance.

## 2. Materials and Methods

### 2.1. Sample Source and Access to NRGs

From the TCGA database (https://portal.gdc.cancer.gov/ (accessed on 12 August 2022)) for EAC transcriptome data and clinical information, a total of 87 EAC samples with complete expression data and survival of ≥30 days were included. The basic information and clinicopathological conditions of the samples are shown in Appendix A. In addition, the verification sample was obtained from the GEO database’s GSE19417 dataset, and a total of 48 EAC samples meeting the requirements were included. From the KEGG (https://www.kegg.jp/kegg/ (accessed on 12 August 2022)) database, we identified 159 necroptosis-related genes; for all of the genes, see Appendix A.

### 2.2. Identification of Differentially Expressed NRGs

The mRNA gene expression data matrix of the EAC samples was obtained by matching the transcription data and human profiles with Perl. Using “FDR < 0.05, |log2FC| > 1” as the standard, the DE-NRGs were extracted by “limma” in R version 4.0.4 software (The R Foundation for Statistical Computing, Vienna, Austria).

### 2.3. Mutation Rate and Functional Enrichment Analysis of the DE-NRGs

The mutation rates of the DE-NRGs were analyzed in the cBioPortal database (http://www.cbioportal.org/ (accessed on 12 August 2022)). The chromosome number and location of 17 DE-NRGs were obtained from the National Center for Biotechnology Information (NCBI) database. Then, the DE-NRGs were analyzed for GO function and KEGG pathway enrichment using the cluster Profiler software package in R. The PPI of the DE-NRGs was further obtained by the STRING (https://cn.string-db.org/ (accessed on 12 August 2022)) database to identify the core genes.

### 2.4. Identification of Prognostic DE-NRGs and Analysis of Tumor Subtypes

The DE-NRGs correlated with the prognosis of EAC were screened by univariate Cox regression (*p* < 0.05). Then, K-means clustering analysis was used to classify EAC samples according to the expression level of the DE-NRGs and to select the K-value that indicated the highest intra-group correlation and the lowest inter-group correlation. Kaplan–Meier (K-M) survival analysis showed the survival of the different subtypes, and heat maps indicated the expression of prognosis-related DE-NRGs in different subtypes, as well as the connection between prognosis-related DE-NRGs and different clinicopathological features.

### 2.5. Establishment and Validation of the Prognostic Model

The LASSO regression algorithm based on the “glmnet” R package was used to screen for the best DE-NRGs associated with EAC prognosis and to build a risk scoring model:$$Riskscore=\overset{n}{\underset{i=1}{\sum }}Coef\left(i\right)*Expr\left(i\right)$$

We calculated the risk scores of each sample and classified the samples into high- and low-risk groups according to the median, and the grouping ability was verified by principal component analysis (PCA). The survival statuses of the two groups were compared by K-M analysis. Univariate and multifactorial Cox regression analyses were used to explore whether the risk score model was a potential independent prognostic indicator for EAC patients. ROC curves were generated to verify the predictive value of the risk score model. The prognostic value of the constructed model was further verified in validation set GSE19417.

### 2.6. GSEA Enrichment Analysis and Immune Activity Analysis

GSEA analysis was performed using GSEA software 4.2.3 (http://www.gsea-msigdb.org/gsea/index.jsp (accessed on 12 August 2022)), and the TOP5 pathways that were significantly enriched between the low- and high-risk groups were selected. Visualization was performed using the “gridExtra, grid, ggplot2” R package. Then, the scores of immune-related functions and immune cells among the different groups were calculated via the ssGSEA algorithm using the R package “reshape2, ggpubr.”

### 2.7. Nomogram and Calibration

The statistically significant clinicopathological characteristics and risk scores in the multifactorial Cox regression analysis were combined to construct a nomogram of EAC patients using the “rms” R package, and the accuracy of the prognostic nomogram was evaluated using the calibration curve.

### 2.8. Statistical Analysis

Statistical analysis was conducted in this study using the R software version 4.0.5 (The R Foundation for Statistical Computing, Vienna, Austria). Differences were considered statistically significant at *p* < 0.05. The DE-NRGs associated with the prognosis of the EAC patients were screened using one-way cox regression. The log-rank statistical method was used for K-M survival analysis. A risk score model was constructed by LASSO regression, its grouping ability was verified by PCA analysis, and the predictive ability of the model was verified by the ROC curve. Univariate and multifactorial Cox regression analyses were used to examine whether the risk score model could be a potential independent prognostic indicator for patients with EAC.

## 3. Results

### 3.1. Identification of 38 DE-NRGs

Based on the TCGA database, we included 9 normal samples and 78 EAC samples for differential analysis. Compared with the normal samples, 38 DE-NRGs were screened by differential analysis, including 3 downregulated genes and 35 upregulated genes. The expression of the 38 DE-NRGs is shown in Figure 1A,B. To understand the mutations of these genes, we examined the mutation rates of the 38 DE-NRGs through the cBioPortal database. As Figure 1C shows, the mutation rate of a total of 17 genes was ≥3%, with deep deletion and gene amplification being the most common types of mutations. Based on the search results of the NCBI database, we found that DE-NRGs were mainly located on chromosomes 1, 4, and 6 (Appendix A).

### 3.2. GO, KEGG, and PPI Analysis of the DE-NRGS

To explore the biological functions and pathways involved in the DE-NRGs, we further conducted enrichment analysis and selected the top 30 pathways and functions of the KEGG and GO enrichment analyses, respectively. As shown in Figure 2A, the KEGG enrichment results indicated that the DE-NRGs were mainly involved in diseases causing necroptosis, as well as the NOD-like receptor signaling pathway, the IL-17 signaling pathway, and the TNF signaling pathway. According to the GO enrichment analysis results, the DE-NRGs were involved in biological processes, including apoptosis, chromatin silencing, and epigenetic regulation (Figure 2B). Then, we explored the interaction between these DE-NRGs’ transcriptional proteins through the PPI protein interaction network. A total of 30 genes were involved in the network (Figure 2C), among which TNF, CASP1, and IL-1B were the core genes (Figure 2D).

### 3.3. Analysis of Tumor Subtypes Based on Prognostic DE-NRGs

The 55 EAC samples with complete prognostic data and clinical case information were included for follow-up analysis. Based on the 38 DE-NRGs, a total of nine genes with prognostic association were detected by univariate Cox regression (Figure 3A). According to the results of the K-means clustering analysis (Appendix A), the 55 EAC samples were classified into two subgroups. Figure 3B shows the expression profiles of the nine prognostic DE-NRGs across the different subgroups and clinical had. The results of the K-M survival analysis revealed that patients with subtype 1 had better survival compared with those with subtype 2 (Figure 3C).

### 3.4. Construction and Evaluation of a Risk Score Model

The nine DE-NRGs with prognostic correlations were further used for LASSO analysis, and four optimal NRGs with prognostic correlations (HMGB1, IL-1B, H2AC12, and H2AC21) were obtained and used to construct a risk scoring model (Appendix A and Figure 4A). The risk score model was as follows: Risk score = 0.386 × Expr _HMGB1_ + 0.435 × Expr _IL-1B_ + 1.810 × Expr _H2AC12_ + 2.914 × Expr _H2AC21_. The EAC patients were separated into high-risk (*n* = 28) and low-risk (*n* = 27) groups via the median risk score. PCA analysis indicated that this model had good discrimination ability (Figure 4B). Further analysis of the survival of patients in the high- and low-risk groups showed that the risk score was negatively correlated with patient survival. In addition, the survival state and risk graphs (Figure 4C–E) revealed the same results.

The ROC curve was then used to estimate the diagnostic power of this prognostic model. The AUC values of this model for one-, two-, and three-year survival were 0.831, 0.913, and 0.886, respectively (Figure 5A). In addition, the risk score model had the highest AUC value of 0.885 compared with the AUC values for age, sex, stage, and TNM stage (Figure 5B). Analysis was performed in verification set GSE19417, and the results were consistent with those of the TCGA database analysis (Figure 6). These results indicate that the model had certain prognostic ability.

### 3.5. Independent Prognostic Value of the Risk Score Model

The effects of risk score and age, gender, stage, and TNM stage on the survival of patients with EAC were analyzed by Cox regression. The results of the univariate Cox regression analysis presented an HR of 2.061 (95% CI = 1.544–2.744) for the risk score (Figure 5C); meanwhile, in the multivariate Cox regression analysis, for the risk score, the HR was 2.051 (95% CI = 1.209–3.479) (Figure 5D). The results showed that, when the clinical confounders were excluded, the risk score remained a risk factor for poor prognosis in EAC patients.

### 3.6. GSEA Analysis

Through GSEA analysis, a total of 178 KEGG pathways (Appendix A) and 5506 GO functions (Appendix A) were identified, and the top five KEGG pathways and GO functions significantly enriched in the two risk groups were selected for visualization, respectively. In the KEGG pathway (Figure 7A), many metabolically associated pathways were significantly activated in high-risk populations, such as the tricarboxylic acid cycle (TCA cycle) and oxidative phosphorylation pathways. The GO enrichment results showed (Figure 7B) that high-risk groups were associated with biosynthesis processes, such as nucleosome assembly and protein localization. In addition, both the KEGG and GO enrichment results indicated that the low-risk group was associated with certain immune-related pathways.

### 3.7. SsGSEA Analysis

We evaluated the differences in the enrichment scores of 13 immune-related functions and 16 immune cells between the two risk groups. IDCs, Mast cells, pDCs, and Tfh had higher immune infiltrative activity in the high-risk group (Figure 8A), while the immune infiltration rates of the other immune cells were not statistically significant between the two groups (*p* > 0.05). Except for type II IFN response immune functioning, which was more significant in the high-risk group (Figure 8B), no obvious differences were seen in the other immune-related functions in both groups (*p* > 0.05) (Figure 8B).

### 3.8. Prognostic Nomogram

Statistically significant stage and risk scores in the multivariate Cox regression were included to construct a nomogram of EAC prognosis. The predicted one-, two-, and three-year survival rates for EAC patients are shown in Figure 9A. This prognostic nomogram had a C-index of 0.792. Based on the calibration curves, we can conclude that the prognostic nomogram constructed in this study was of good agreement (Figure 9B–D).

## 4. Discussion

Early detection of EAC is particularly important because of the late onset and aggressive nature of clinical symptoms in patients with EAC. Necroptosis, a novel mode of cell death, plays a key role in the regulation of cancer biology, including tumorigenesis, cancer metastasis, immunity, and subtypes [16,17], and its dual effect on tumors has been confirmed [18,19]. Considering the biological role of necroptosis factors, we can use them as a potential biomarker for cancer treatment. In this study, a prognostic risk model based on four DE-NRGS was successfully constructed, which could better predict the prognosis of EAC patients. Moreover, the GSEA and ssGSEA results revealed that necroptosis genes may be associated with the progression of EAC by changing metabolic phenotypes or regulating tumor immunity. In addition, the nomogram constructed by the combined analysis of the risk score model and clinicopathological features could predict the prognosis of EAC well, with a C-index of 0.792, indicating that the prognostic risk model based on the four DE-NRGs was useful for predicting the prognosis of EAC as a prognostic predictive biomarker.

In this study, 38 DE-NRGs were first identified based on TCGA transcriptomic data and included in the GO, KEGG, and PPI analyses. The results showed that these DE-NRGs were significantly enriched in diseases, regulatory pathways, and other processes related to necroptosis. Many studies have demonstrated that NLRs play a key role in tumorigenesis, angiogenic metastasis, cancer cell dryness, and chemotherapy resistance. Tumor progression can be inhibited by blocking the signaling pathway of NLRs with various plant drugs, and the synthesis of small molecules and microRNA [20]. Ma et al. [21] revealed that the knockdown or overexpression of SPRY4-IT1 can lead to changes in protein levels within the TNF signaling pathway, suggesting that the long non-coding RNA SPRY4-IT1-mediated TNF signaling pathway could be a therapeutic target for hepatocellular carcinoma. Another study also suggested that miR-383 may target IL-17 through the STAT3 signaling pathway and thus exert antitumor effects in hepatocellular carcinoma [22]. Combined with the above studies, we can conclude that necroptosis factors may be involved in the occurrence and progression of EAC through these signal transduction pathways.

To determine the best DE-NRGs associated with EAC prognosis, four NRGs (HMGB1, IL-1B, H2AC12, and H2AC21) were screened for risk model construction. The expression results showed that these four NRGs were highly expressed in the high-risk group, suggesting that these genes may be related to the tumor progression of EAC patients and may be oncogenic genes. Some of the NRGs used in the risk model have been confirmed to play an important role in EAC. Previous studies have revealed that HMGB1 creates an inflammatory tumor microenvironment between EAC cells and macrophages, which further promotes the progression of EAC [23]. IL-1B is considered to be the pivotal regulator of tumor progression, metastasis, and immunosuppression [24,25]. In this study, IL-1B was also the core gene of the PPI protein interaction network. Fei et al. [26] reported that IL-1B is a risk immune-related differential gene of EAC and participates in the related biological process of EAC. However, the roles of H2AC12 and H2AC21 genes in tumors have not been explored so far; therefore, the mechanisms of these genes in tumor development can be further investigated in the future. The results of the survival analysis showed that the low-risk group had a better survival rate and a longer survival time. Moreover, the prognostic risk model was proven to have a certain prognostic prediction ability and could be used as an independent prognostic indicator of EAC.

In addition, we explored the enriched pathways and functional characteristics of different risk populations through GSEA. Some metabolism-related pathways, such as the TCA pathway, were remarkably enriched in the high-risk population, suggesting that necroptosis may affect the metabolic phenotype of the organism, which, in turn, mediates important biological processes in the organism. Studies have shown that tumor cells harness the TCA cycle in a different way to normal cells, resulting in tumor cells that are more sensitive to suppressors that target reprogrammed metabolic pathways in the TCA cycle [27]. Based on this idea, targeting the TCA cycle by circulating enzymes or small molecule inhibitors that regulate circulating enzymes could be an effective cancer treatment. The GO results showed that the high-risk group was related to the biosynthesis process. Therefore, necroptosis can effectively promote the metabolism, accelerate the cell cycle, and promote biosynthesis. We further investigated the differences in immune cells and immune activity in the two populations. Immune cells (IDCs, Mast cells, pDCs, and Tfh) and immune pathways (type II IFN response) were active in the low-risk population, and some of these cells and pathways were closely associated with necrotizing apoptosis. Some studies have revealed that necroptosis factors can provide inflammatory cytokines and tumor-specific antigens for dendritic cell maturation, which, in turn, induce immune stimulation and the cross-activation of CD8^+^ T cells [28]. These results demonstrate that necroptosis may promote or reduce the progression of EAC by mediating tumor immune-related processes.

However, this study has some limitations. First, we could not obtain sufficient external datasets that met the conditions to verify the conclusions of this study. Second, although this study proposed possible targets and mechanisms of necroptosis factors in the occurrence and progression of EAC, no experiments have been conducted to further verify this theory. Therefore, further studies are needed to confirm these conclusions.

## 5. Conclusions

The prognostic risk model based on the four DE-NRGs constructed in this study showed more accurate predictive efficacy in predicting the prognosis of EAC patients, and the related functional analysis helped to elucidate the mechanism of the role of necroptosis factors in the development of EAC. Taken together, the findings of our study may provide new strategies for finding new therapeutic targets and prognostic indicators for EAC.

## Figures and Tables

**Figure 1 genes-13-02243-f001:**
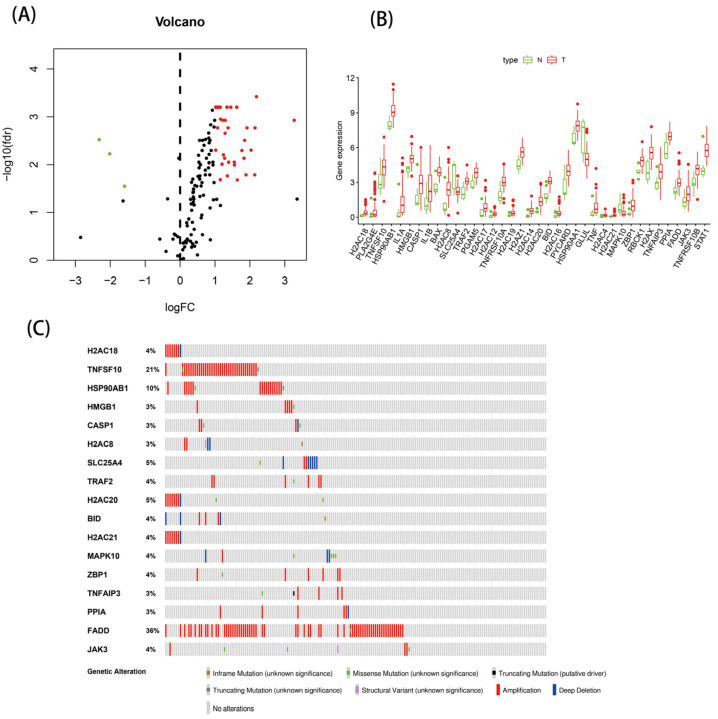
The expressions of 38 DE-NRGs: (**A**) heatmap; (**B**) boxplot; and (**C**) 17 DE-NRGs with a mutation rate of ≥3%.

**Figure 2 genes-13-02243-f002:**
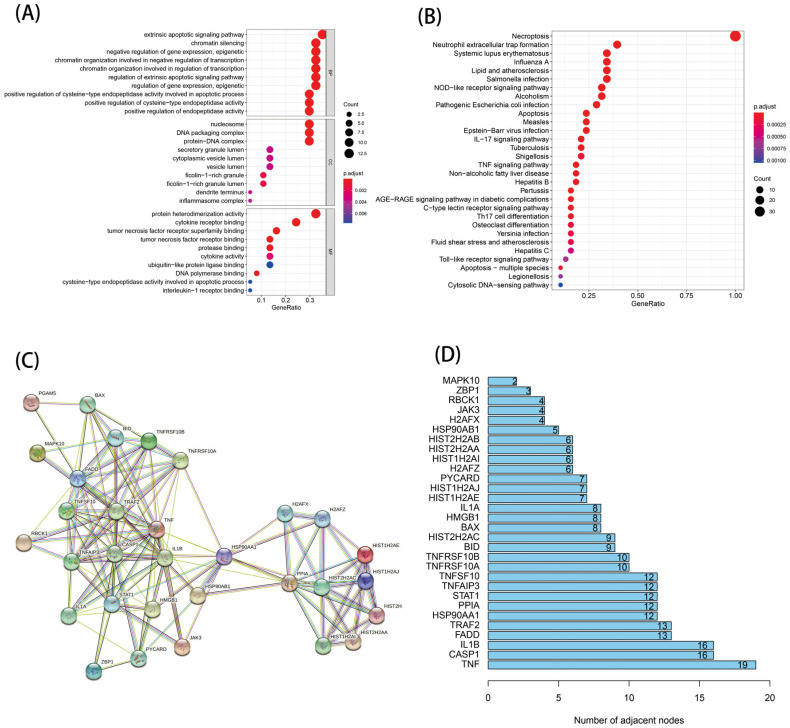
Function analysis of the 38 DE-NRGs and their interactions: (**A**) GO analysis; (**B**) KEGG analysis; and (**C**,**D**) PPI analysis.

**Figure 3 genes-13-02243-f003:**
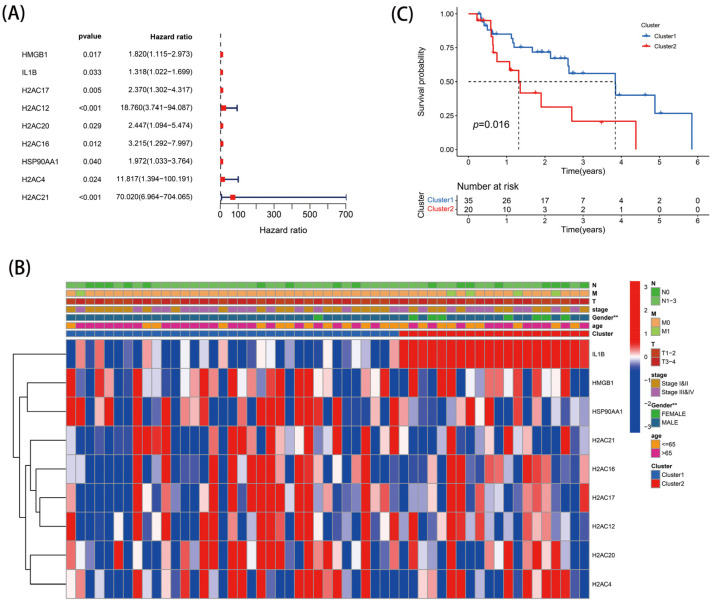
Tumor classification: (**A**) DE-NRGs relevant to the prognosis of EAC patients; (**B**) heatmap; and (**C**) Kaplan–Meier OS curves. ** *p* < 0.01.

**Figure 4 genes-13-02243-f004:**
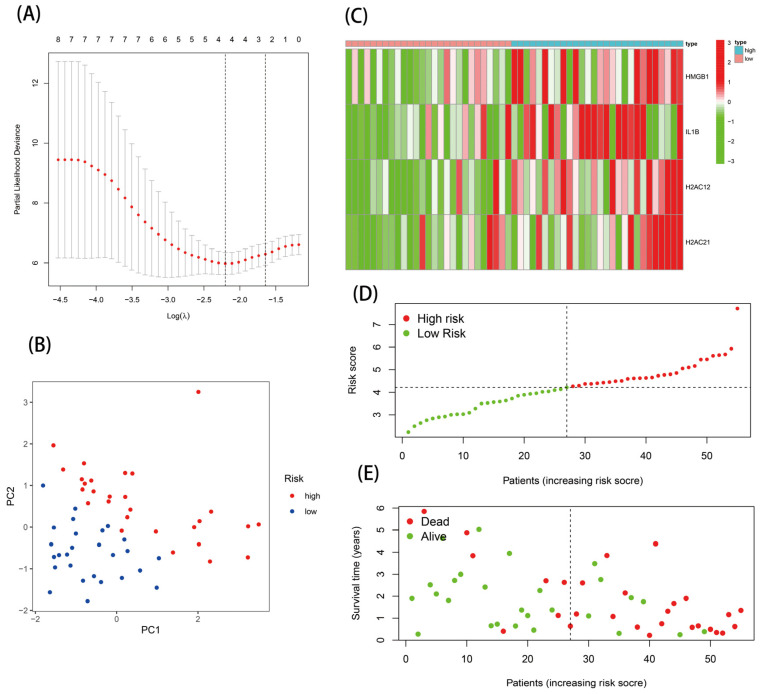
Establishment of the risk score model: (**A**) four prognostic DE-NRGs were identified; (**B**) PCA analysis; (**C**) heatmap; (**D**,**E**) risk survival status plot.

**Figure 5 genes-13-02243-f005:**
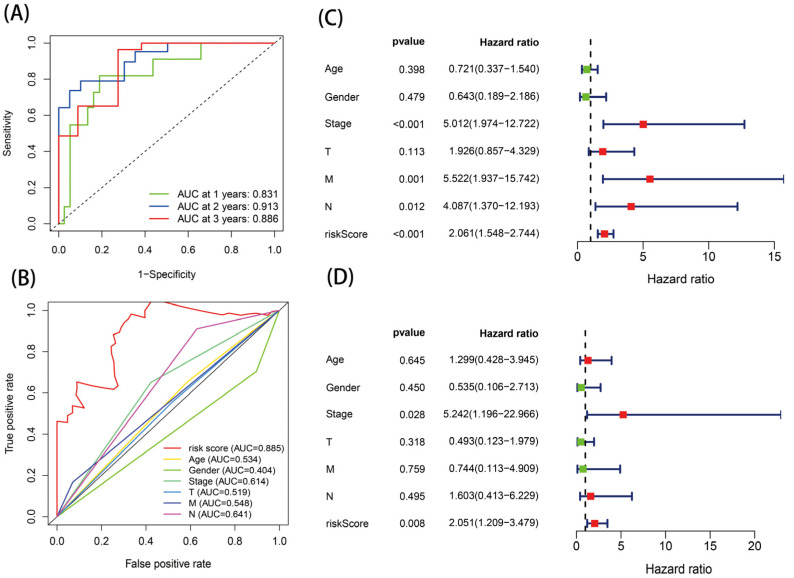
Identification of the risk score model: (**A**,**B**) ROC analysis and (**C**,**D**) independent prognostic analysis via univariate and multivariate cox regression analysis.

**Figure 6 genes-13-02243-f006:**
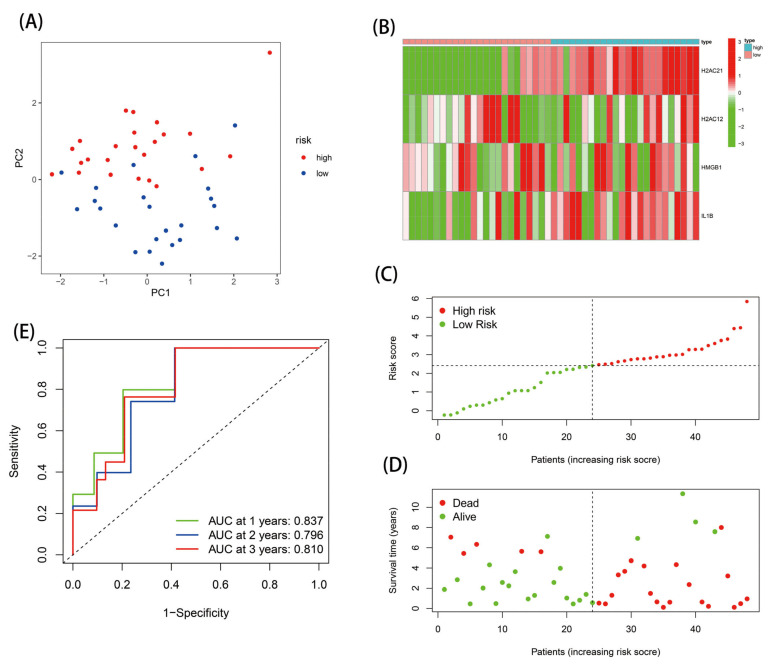
Validation of the risk score model in the GEO database: (**A**) PCA analysis; (**B**) heatmap; (**C**,**D**) risk survival status plot; and (**E**) ROC analysis.

**Figure 7 genes-13-02243-f007:**
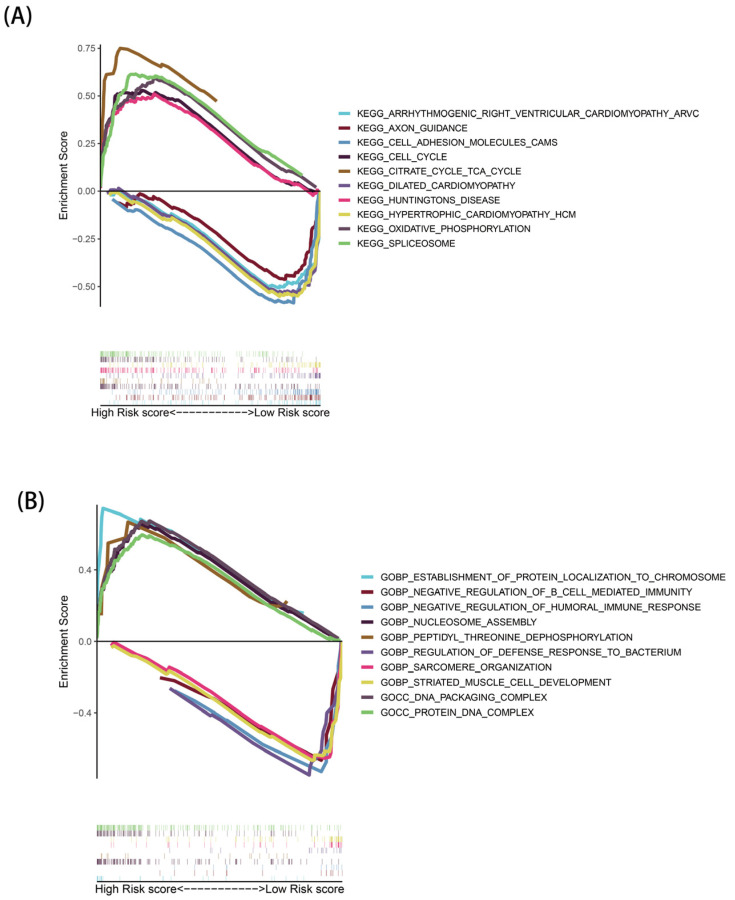
Gene set enrichment analysis: (**A**) KEGG analysis and (**B**) GO analysis.

**Figure 8 genes-13-02243-f008:**
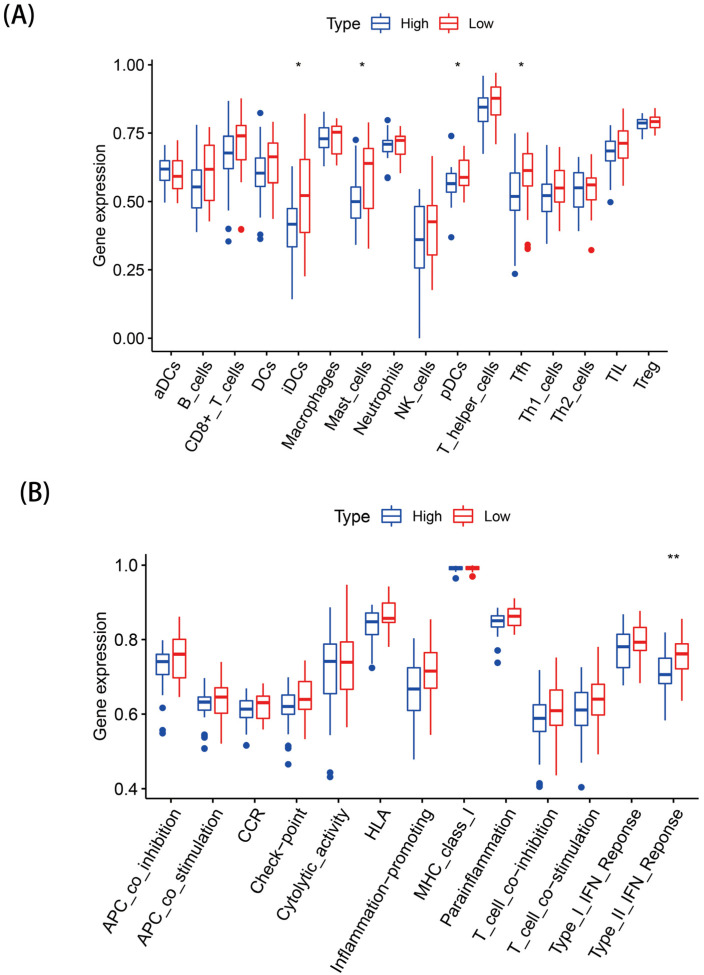
Landscape of immune infiltration in the two groups: (**A**) Differential analysis of the infiltration of 22 immune cells and (**B**) immune functional differences. * *p* < 0.05; ** *p* < 0.01.

**Figure 9 genes-13-02243-f009:**
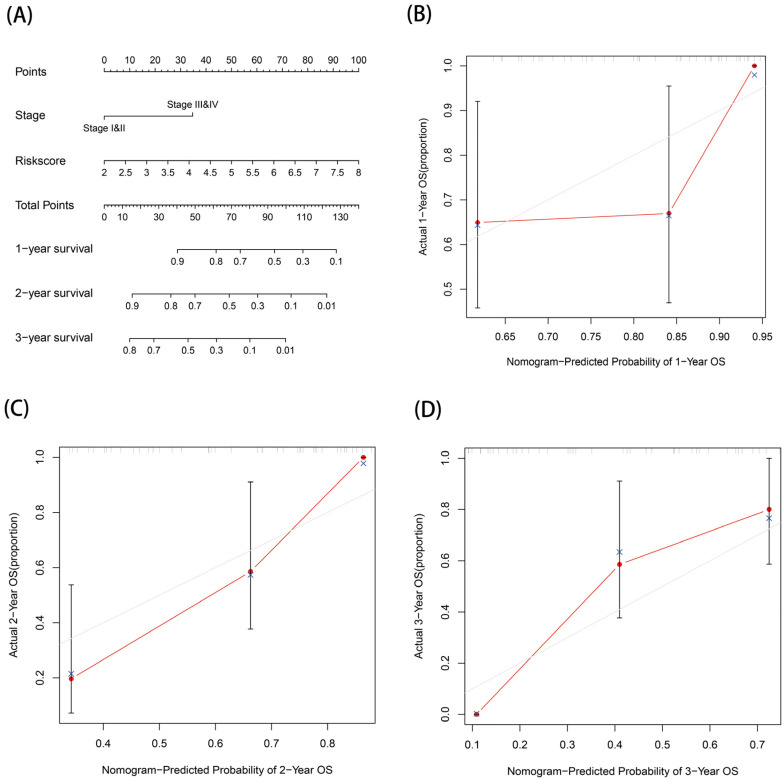
Establishment and validation of a nomogram for prognostic prediction: (**A**) nomogram and (**B**–**D**) calibration curves for 1-, 2-, and 3-year OS of the nomogram.

## Data Availability

The datasets used in this study can be found in the online repository. The serial numbers of the datasets used are provided in this article.

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
