# Peer review of "Development and Validation of a Prognostic Model for Esophageal Adenocarcinoma Based on Necroptosis-Related Genes"

_genes, 2022, doi:10.3390/genes13122243_

Round 1
Reviewer 1 Report
Dear Authors
The research article entitled “Development and Validation of a Prognostic Model for Esophageal Adenocarcinoma Based on Necroptosis-Related Genes” by Zhang et al is well written one and novel in association with EAC since the validation of prognostic model of ESCC is well established through lot of publications. In this research article, the authors tried to develop and validate the prognostic model for esophageal adenocarcinoma based on necroptotic genes. The authors tried to validate necroptic genes for the prognosis of EAC through various computational analyses and represented through his figures. However, the quality of the figures is very poor for the readers as well as for the reviewers to come to any conclusion. The author needs to thoroughly revise and check the manuscript and resubmit with the clear figures and corrections.
Comments to the authors:
1. Can authors explain the rationale for selecting only the 9 Normal tissues when compared to 78 EAC tissues in TCGA database.
2. Fig 1B and 1C, need to be replace with the high-quality and resolution figures since it is tough to analyze the figures.
3. Fig 1D is missing, “Figure 1D, there were 17 genes with a mutation rate ≥3%, among 151 which gene amplification and deep deletion were the most common mutation types”
4. The quality of Fig 2A and 2B is also not clear to read.
5. Fig 3B is not clear, need to be replaced.
If If possible the authors can check the basal level and expression of NEG genes in ESCC, Barrett's (Low grade and high grade dysplatic cells) and EAC cells such as OE33, OE19, FLO1 and SKGT4. it will give more solid conclusions rather than the computational analysis.
Author Response
Response to Reviewer 1 Comments
Point 1: Moderate English changes required.
Response 1: We thank the reviewer’s suggestion. We polished the language by the editing service to improve readability of the manuscript.
Point 2: However, the quality of the figures is very poor for the readers as well as for the reviewers to come to any conclusion. The author needs to thoroughly revise and check the manuscript and resubmit with the clear figures and corrections.
Response 2: We thank the reviewer for this important comment and agree that the original version of the manuscript did not provide the clear pictures. We reprocessed the resulting figures to ensure that each image was at high resolution and changed the images in the paper manuscript.
Point 3: Can authors explain the rationale for selecting only the 9 Normal tissues when compared to 78 EAC tissues in TCGA database.
Response 3: Thank you for your questions. The sample data in this study came from the TCGA database, which collected a variety of data from more than 20,000 samples of 33 types of cancer, including transcriptome expression data, genomic variation data, methylation data, clinical data, etc. The source of tissue samples was tumor tissue and normal tissue samples voluntarily donated by cancer patients. A total of 78 EAC tissue samples and 9 normal tissue samples were collected in the database. We included all normal tissue samples for the study.
Point 4: Fig 1B and 1C, need to be replace with the high-quality and resolution figures since it is tough to analyze the figures; The quality of Fig 2A and 2B is also not clear to read; Fig 3B is not clear, need to be replaced.
Response 4: We are sorry that we did not present clear and high quality figures before. Now we have found a better method to preserve high resolution of our figures, and we believe that the new figures are satisfactory in quality and could display clear details after enlargement.
Point 5: Fig 1D is missing, “Figure 1D, there were 17 genes with a mutation rate ≥3%, among 151 which gene amplification and deep deletion were the most common mutation types”.
Response 5: We apologize for the error in the earlier manuscript. Figure 1D is actually Figure 1C. Due to our carelessness, there were mistakes in the previous manuscript. This error has been corrected in the current manuscript.
Point 6: If possible the authors can check the basal level and expression of NEG genes in ESCC, Barrett's (Low grade and high grade dysplatic cells) and EAC cells such as OE33, OE19, FLO1 and SKGT4. it will give more solid conclusions rather than the computational analysis.
Response 6: We agree that it may be more meaningful to detect the basal level and expression of NEGs gene in ESCC, Barrett(low grade, high grade dysplasia cells), and EAC cells. In fact, this is the future research direction that our laboratory is interested in, and our follow-up research plan is to verify it through various experiments. We will conduct relevant experimental research in the future.
Reviewer 2 Report
In this manuscript, the authors integrated the expression profile of 9 necroptosis‐related genes (NRGs) to develop a novel prognostic model for Esophageal adenocarcinoma. However, the study lacks in validation in least one independent cohort and is therefore not convincing. Moreover, some of the descriptions of the manuscript are not clear.
Major points:
1. In line 151, the authors mentioned that gene amplification and deep deletion were the most common mutation types. As we knew EAC is mainly driven by chromosomal instability [1], it would be much helpful to provide the genomic loci of these gene and determine if there is a cluster of NRGs in a region of the human genome.
2. In line 175, more details about the K-means clustering analysis to divide EAC samples into two groups should be addressed in the method part. In addition, fiugre 3B-D, Figure 4A are all about the methodology and should be moved into supplementary data.
3. For figure 4C, 3-D PCA plot is a fascinating way to visualize the data, but sometimes is not good way to display the distance between two groups. Here a 2-D PCA plot could be an option in terms of optimizing the data visualization.
[1] Liu, Y. Cancer Cell. 2018 Apr 9; 33(4): 721–735.e8.
Author Response
Response to Reviewer 2 Comments
Point 1: Extensive editing of English language and style required.
Response 1: We apologize for the poor language of our manuscript. We worked on the manuscript for a long time and the repeated addition and removal of sentences and sections obviously led to poor readability. We have now worked on both language and readability and polished the language by the editing service. We really hope that the flow and language level have been substantially improved.
Point 2: However, the study lacks in validation in least one independent cohort and is therefore not convincing.
Response 2: We thank the reviewer for pointing out this issue. We indeed should verify our conclusions in the independent cohort. By querying the GEO database and searching for EAC data that meets the requirements, we finally selected a dataset that meets the requirements most to verify our research. In the revised manuscript, a validation of the prognostic value of risk score model has been added (Figure 6). The relevant descriptions in the manuscript are found in lines 102-104 and 285-287.
Point 3: In line 151, the authors mentioned that gene amplification and deep deletion were the most common mutation types. As we knew EAC is mainly driven by chromosomal instability [1], it would be much helpful to provide the genomic loci of these gene and determine if there is a cluster of NRGs in a region of the human genome.
Response 3: We agree with this recommendation and add to the analysis in this section. The results of the analysis are supplemented by Supplemental Table S3. The relevant descriptions in the manuscript are found in lines 133-134 and 197-199. We will be happy to edit the manuscrip further, based on helpful comments from the reviewer.
Point 4: In line 175, more details about the K-means clustering analysis to divide EAC samples into two groups should be addressed in the method part. In addition, fiugre 3B-D, Figure 4A are all about the methodology and should be moved into supplementary data.
Response 4: The statements have been corrected. The details about the K-means clustering analysis to divide EAC samples into two groups have been added to the methods section of our revised manuscript, and fiugre 3B-D, Figure 4A also have been added to the Supplemental Figure S1.
Point 5: For figure 4C, 3-D PCA plot is a fascinating way to visualize the data, but sometimes is not good way to display the distance between two groups. Here a 2-D PCA plot could be an option in terms of optimizing the data visualization.
Response 5: We agree with this suggestion and have modified the presentation of the image, as shown in Figure 4B.
Round 2
Reviewer 1 Report
Dear Authors
I appreciate for the effort made by the authors to correct the manuscript. However, I cannot able to make any conclusion, if the figures are not clearly represented.
Reviewer 2 Report
All my questions have been addressed well. Thanks!